# Detection of Hepatitis E Virus (HEV) in Pigs and in the Wild Boar (*Sus scrofa*) Population of Chieti Province, Abruzzo Region, Italy

Fabrizio De Massis [1], Giuseppe Aprea [1], Silvia Scattolini [1,*], Daniela D'Angelantonio [1], Alexandra Chiaverini [1], Iolanda Mangone [1], Margherita Perilli [1], Giulia Colacicco [1,*], Sabrina Olivieri [1], Francesco Pomilio [1], Adriano Di Pasquale [1], Giacomo Migliorati [1], Giovanni Di Paolo [2], Chiara Morgani [2] and Angelo Giammarino [2]

1  Istituto Zooprofilattico Sperimentale dell'Abruzzo e del Molise "G. Caporale", 64100 Teramo, Italy
2  Azienda Sanitaria Locale Lanciano Vasto Chieti, Via Martiri Lancianesi, 17/19, 66100 Chieti, Italy
*  Correspondence: s.scattolini@izs.it (S.S.); giulia.colacicco97@gmail.com (G.C.)

**Abstract:** Hepatitis E virus (HEV) is a zoonotic pathogen, causing infectious hepatitis in man. Pigs and wild boars are the natural asymptomatic reservoirs, while the disease in humans could be either asymptomatic or evolve in hepatitis. In Europe, an increasing number of human infections from HEV have been reported over the last few years. The main route of transmission is through contaminated food, by direct or indirect consumption of raw or undercooked pork and wild boar meat and meat products. Up to now, HEV prevalence in Italian northern regions has been extensively determined in wild boars and pigs, while less data have been collected from the southern ones. There is a need to report more data about HEV prevalence from wild boars and pigs in southern Italy in consideration of the potential risk posed by some specific traditional food products manufactured in these areas and produced from pig and wild boar livers (e.g., sausages and salami). The aim of this study was to demonstrate the circulation of the Hepatitis E virus (HEV) in pigs and in the wild boar population of the province of Chieti, Abruzzo Region, Central Italy. Moreover, potential HEV seroprevalence in hunters from that area was also assessed. The overall prevalence of HEV RNA in wild boars was 9.5% (CI 5.4–16.2%), but no HEV RNA was detected in samples from pigs.

**Keywords:** wild boar; pig; epidemiology; foodborne disease; hepatitis E virus (HEV); public health; surveillance; wildlife; zoonoses

## 1. Introduction

Hepatitis E virus (HEV) is one of the viruses responsible for hepatitis in humans worldwide in different geographical areas [1]. Antibodies against HEV have been identified in different wild and domestic animal species [2,3]. HEV is clearly identified as a zoonotic agent [4–8], and the circulation of the same HEV subtypes between animals and humans has been extensively demonstrated [9,10]. Experimental clinical trials are underway for the marketing of vaccines. In European Union Member Countries, over the last 10 years, more than 21,000 acute clinical cases with 28 fatalities have been notified, with an overall 10-fold increase in reported HEV cases [11]. In particular, the number of human laboratory-confirmed cases of hepatitis E in Europe has dramatically increased in recent years, from 514 in 2005 to 5617 cases in 2015. In total, 28 deaths associated with HEV infection were reported from five countries between 2005 and 2015 [12]. In developing countries, HEV is transmitted mainly via the faecal–oral route through the consumption of contaminated water, while in industrialised countries, most cases have been related to the consumption of meat (deer and wild boar) or meat products (salami and sausages from pig/boar meat) eaten raw [11]. Several cases of human Hepatitis E were found in Abruzzo, mostly in the

Province of L'Aquila, and mainly related to consumption of wild boar/pig meat and liver products, such as dried sausages, eaten raw or lightly cooked [13].

HEV is classified into four genotypes. Genotype 1 (g1) and g2 cause infections in humans in developing countries, while g3 and g4 are considered zoonotic, and pigs and wild boars are recognised as the main reservoir. Many authors have reported evidence of direct transmission of HEV g3 and g4 from animals to humans through the consumption of undercooked meat [14]. The mortality from HEV in human beings is generally low (less than 1%), but disease severity has been reported in 28% of pregnant women, and the genotype involved is, according to the case, only Genotype 1. Actually, HEV is also responsible for neurological manifestations such as Guillain-Barré syndrome, neuralgic amiotrophy (NA) and meningoencephalitis. The cases studied concern Genotype 1 in Asia and Genotype 3 in Europe. The same genotypes are also involved in thrombocytopenia and membranous glomerulonephritis. In India, a lot of cases of acute pancreatitis have been associated with Genotype 1, which is also responsible for different patterns of anaemia. Further, Genotype 3 in immunosuppressed and haematolagical patients can cause chronic hepatitis and cirrhosis induced by liver fibrosis [15]. In Italy, most of the data about prevalence of HEV in wild and domestic animals are limited to central-northern regions, where g3 is the most commonly found with the related subtypes 3e and 3f [16]. Less data about the circulation of HEV are available for pigs, wild boars and humans from Italian southern regions, as well as for the genotypes and subtypes involved. The aim of this study was to estimate the prevalence of HEV in pigs and in the wild boar population of Chieti province (Abruzzo Region, Central Italy) in order to provide more data supporting an evaluation of the overall national prevalence.

## 2. Materials and Methods

### 2.1. Sample Collection

The study was carried out from 2015 to 2019 on the domestic and wild *suidae* population of the Province of Chieti (Abruzzo Region, Central Italy) in collaboration between the Istituto Zooprofilattico Sperimentale dell'Abruzzo e del Molise "G. Caporale" (IZS-Teramo) and the Veterinary Services of the Local Health Unit Lanciano-Vasto-Chieti (LHU). This choice was suggested by the consideration that this area is the one with the highest concentration of free-living wild boars of the entire Abruzzo region. The geographical area under study coincides then with the province of Chieti, which has an extension of 2588.35 km$^2$ (of these, 1080.7 km$^2$ represent agro-forestry-pastoral territory), mainly characterised by the presence of hills, foothills and mountains and which shares more than a quarter of its borders with the Majella National Park. The wild boars in the area were estimated in around 28,000 units at the time of sampling. A representative sample of wild boars to be tested was identified with classical statistical techniques, i.e., a sample of 116 individuals were selected, considering an expected prevalence of 15% and a 95% confidence level. Hunters were made aware of the aim of the projects through several meetings with the LHU and were trained for safe sampling of the hunted wild boars. After that, they were provided with the needed tools for sampling, i.e., personal protective equipment (gloves, mask and disposable gown), adequate materials for sample collection and a form to be filled with animal identification data. After each hunting session, the trained hunter proceeded to sample collection, immediately after evisceration on site. From each carcass, a sample of liver (20–30 g) and the entire gallbladder (including the bile) were collected for virus RNA detection. Samples were put into sterile containers, immediately refrigerated and dispatched to IZS-Teramo at a temperature of 4 ± 2 °C.

The same sampling procedure was carried out in pigs slaughtered in selected slaughterhouses in the area under study. Liver and gallbladder samples were collected from pigs coming from closed farms or kept free-ranging, all of which farmed in structures located on the territory under study. Here also, samples were put into sterile containers, immediately refrigerated and sent to laboratories of IZS-Teramo at the temperature of 4 ± 2 °C.

### 2.2. HEV Genome Detection

The analysis from livers and gallbladders were performed on 1 ± 0.2 g of sample adding phosphate buffered saline (PBS), and quartz fine granules for viral concentration RNA was extracted through the Nuclisens magnetic extraction kit (Biomerieux, Marcy-l'Etoile, France) and the Nuclisens MiniMag platform, following the protocol described by G. Aprea et al., 2020 [17]

The detection of the virus genome and the analyses of the results were carried out following the protocol by Di Bartolo et al., 2012 [18], using primers and probes described by Jothikumar et al., 2006 [19] and Martínez-Martínez et al., 2011 [18–20].

### 2.3. Whole Genome Sequencing (WGS)

RNA samples that tested positive with a Ct value < 25 undertook the whole genome sequencing process following the protocol described by Lorusso et al., 2022 [21]. Purification was performed using ExpinTM PCR SV (GeneAll Biotechnology CO., Seoul, Korea) and quantification through the QuantiFluor One ds DNA System kit (Promega, Madison, WT, USA). Nextera DNA Flex Library Prep (Illumina Inc., San Diego, CA, USA) was used for the preparation of libraries according to the manufacturer's protocol. MiniSeq Mid Output Kit (300-cycles) and standard 150 bp paired-end reads were used for deep sequencing on the MiniSeq (Illumina Inc., San Diego, CA, USA).

### 2.4. WGS Data Analysis

An in-house pipeline was used for the analysis of WGS data [22], including trimming (Trimmomatic v0.36) and quality control check of the reads (FastQC v0.11.5). SPA desv3.11.1 was utilised for the de novo Genome assembly of paired-end reads [23]. ABRicate was performed, after genome assemblies, for the de novo filtering for the scaffolds with a minimum length of 200 nucleotides and for matching the best reference to each assembly [24]. In conclusion, the mapping with references identified in the previous step using Bowtie2 (v.2.1.0) was performed [25]. The genotype and subtype of the obtained draft genome was calculated querying the Hepatitis E Virus Genotyping Tool (https://www.rivm.nl/mpf/typingtool/hev/, accessed on 14 September 2022).

## 3. Results

In total, the expected 116 wild boars were sampled. RNA from HEV was detected in 11 individuals (9.5%, CI 5.4–16.2%) (Table 1). Positivity found in the gallbladder only (6) was greater than in the liver only (3) or in both (6). Considering an estimated population of about 28,000 wild boar units, the number of estimated infected heads would be 2269, ranging from 1303 to 3910, according to the confidence intervals. The number of pigs tested for HEV was 117. The virus was not detected either in pigs farmed in closed farms or in free-ranging pigs (0.0%, CI 0.0 & −3.1%). The animals sampled, their geographical origin and the results of the HEV genome identification tests are shown in Figure 1.

**Table 1.** Prevalence of Hepatitis E Virus (HEV) in pigs and wild boars according to different reports from studies published in the international literature.

| Places | Percent of Positive Pigs | Percent of Positive Wild Boars | Reference |
| --- | --- | --- | --- |
| Lazio–Prov. Latina e Frosinone | na | 40.7% (93/228)-[CI] 34.4–47.1% | [26] |
| Australia | 25.4%, (15/59) [CI] 16.1–37.9% | na | [24] |
| Japan | 58% (1448/2500) | 9.0–27.1% | [27] |
| Lazio–Prov. Viterbo | na | 16.3% [CI] 12.7–20.6% | [28] |
| Calabria | na | 26.7% (28/86)-[CI] 18.5–37.0% | [1] |
| Campania | na | 12.36% (11/88)-[CI] 7.2–21.0% | [1] |

**Table 1.** *Cont.*

| Places | Percent of Positive Pigs | Percent of Positive Wild Boars | Reference |
|---|---|---|---|
| Abruzzo | na | 20.14%; (29/144)-[CI] 14.4–27.4% | |
| Abruzzo–Prov. Chieti | 0.0%-[CI] 0.0–3.1% | 9.5%-[CI] 5.4–16.2% | |
| Abruzzo–Prov. Teramo | na | 20.1% | |

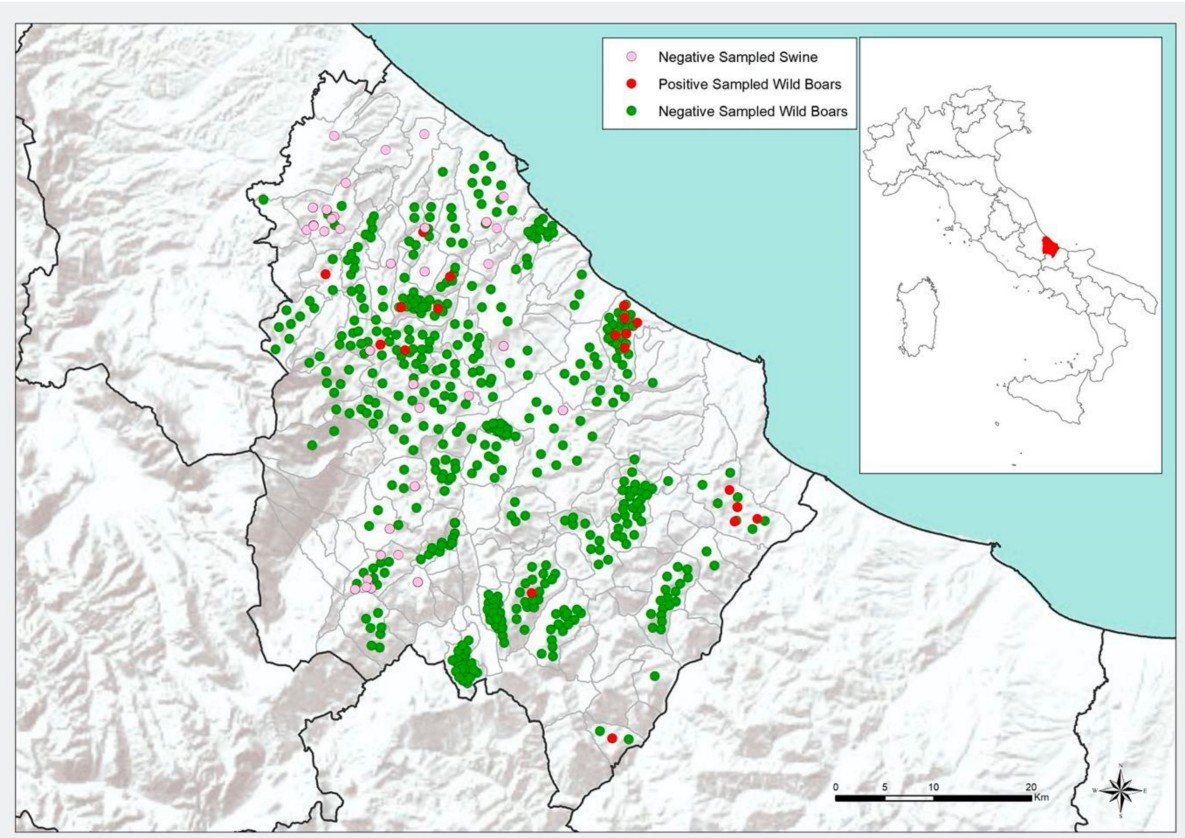

**Figure 1.** Wild boars and pigs sampled between 2015 and 2019 in the Chieti province, their geographical origin and the results of the HEV genome identification tests.

The analysis carried out for HEV WGS gave conclusive results only for one sample. The total length of the draft genome generated from the one HEV positive RNA sample (2019.PE.8890.1.3) and mapped against the reference genome KU176129 was 7140 bp. Querying the HEVnet tool database, the draft genome showed to belong to genotype 3c and was submitted to the National Center for Biotechnology Information (NCBI) under the following accession number: OK665855. In order to verify the relatedness occurring among HEV strains circulating in Abruzzo, we clustered the HEV sequence obtained in this study with the HEV 3c Abruzzo sequences previously published by Aprea et al., 2020 [17] using the Geneious Prime software [29]. The results showed that 2019.PE.8890.1.3 clustered with HEV sequences obtained in wild boars kept in L'Aquila province (2018.AZ.6050.8.2, 2018.AZ.6050.8.7, 2018.AZ.6050.8.10 and 2018.AZ.6050.8.11). (Figure 2).

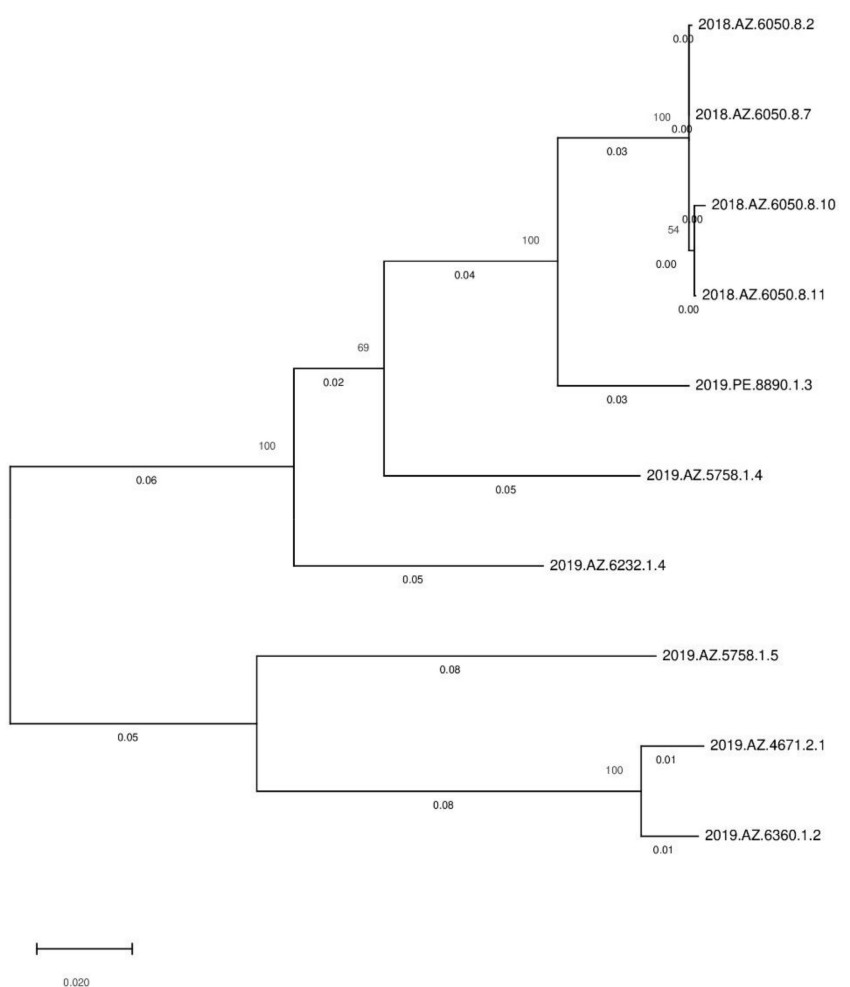

**Figure 2.** Phylogenetic tree shows relationship between the isolate described and the HEV 3c Abruzzo sequences previously published (It has been modified from Aprea et al., 2020 [17]).

## 4. Discussion and Conclusions

Monitoring the health of wildlife is a fundamental element for the proper management of natural ecosystems and for public health. Indeed, the health management of the livestock sector alone cannot guarantee the health of a territory; therefore, an organic control system including wildlife is necessary. Actually, the health status of wild animals is not easily assessable or identifiable with clinical examination. In fact, they live in large areas and have total freedom of movement, making it often impossible to identify the sick animals as it can be done in closed domestic farming.

In Italy, because the wildlife may represent a zoonotic reservoir for various pathogens, each region implements its own guidelines for epidemiological surveillance. However, to date, only few regions have inserted HEV in the list of pathogens under surveillance, especially regions from Northern Italy, where HEV diffusion has been extensively studied in both wild boar and pig farms [16,30,31].

On the contrary, only few studies have been carried out on the wild and domestic *suidae* population in central and southern Italy. In a study conducted in Lazio, between the provinces of Latina and Frosinone, 228 wild boars were sampled and serologically tested for anti-HEV antibodies during the hunting season 2010–2011. In 164 out of the 228 sampled animals, the authors collected the liver parenchyma. In addition, 20 samples of blood and serum from hunters were analysed. The estimated HEV seroprevalence in wild boars and in hunters was 40.7% (93/228; 95% confidence interval (CI) 34.4–47.1%) and 25% (5/20; CI

6.1–43.9%), respectively. A total of 55 of 164 tested wild boar liver samples (33.5%; 95% CI 26.2–40.7%) and 3 of 20 (15.0%; 95% CI 1.3–28.7%) tested human blood samples were positive for HEV RNA PCR. Phylogenetic analysis of the nucleotide sequences obtained from PCR products indicated that the HEV strains present in wild boars and the human population all belonged to genotype 3, supporting the zoonotic role of wild boars in the spread of HEV infection [26]. These are significantly higher percentages than those reported in the literature even in Australia for wild pigs (15 out of 59 tested, 25.4%, CI 16.1–37.9%) or Japan (9.0–27.1%); [27].

Another study, conducted in Abruzzo, Campania and Calabria regions, in collaboration with the Velino-Sirente Regional Park, examined 291 wild boar liver samples taken in protected natural areas. These samples were analysed by real-time RT-PCR, and a HEV RNA prevalence of 13.7% (CI 10.3–18.2%) was reported [1].

Another study assessed HEV occurrence, viral load and genetic variability in wild boars hunted for domestic consumption in the district of Viterbo (Central Italy), where high anti-HEV IgG seroprevalence values were reported in humans. A total of 332 livers and 69 intestine samples were obtained from wild boars hunted between 2011 and 2014. The liver tissue from 54 animals (16.3%, CI 12.7–20.6%) resulted positive for HEV RNA detection. Twenty-six samples were characterised as genotype 3 with the following four subtypes identified: 3a, 3c, 3f and 3l [28](Table 1).

Presence of HEV RNA was also investigated by real-time RT-PCR by De Sabato et al. (2019) [32] in paired livers and muscle samples collected from 196 wild boars hunted in two areas of Central and Southern Italy. Twenty animals (10.2%, CI 6.7–15.2%) tested positive for HEV RNA detection in livers, 11 of which tested positive also in muscles (5.6%, CI 3.2–9.8%). Phylogenetic analyses confirmed the circulation of HEV g3 strains related to the following subtypes: 3c, 3f and 3i. Moreover, some strains were not assigned to any known subtype, suggesting the possibility about the circulation of new types [32].

Recently, an assessment of HEV occurrence and genetic variability in Calabrian wild boars hunted in the central and iconic area of Catanzaro's province, southern Italy, has been performed. A total of 86 wild boar liver samples were analysed showing an overall HEV RNA prevalence of 26.7% (23/86, CI 18.5%–37.0%). All positive RNA samples belonged to HEV g3c subtype [21].

To the best of our knowledge, all studies carried out in central and southern Italy confirm the circulation of the HEV among wild boars, demonstrating how this animal species may act as a potential reservoir of disease for both domestic pigs and humans.

The present study also demonstrates the circulation of HEV in the wild boar population of the Chieti province, Abruzzo region, southern Italy. Even though in our study the HEV genome detection in pig livers gave negative results, the potential circulation of HEV in local pigs cannot be neglected. The prevalence of HEV in the wild boar population in the Chieti province (9.5%, CI 5.4–16.2%) resulted lower than the prevalence reported from Aprea et al. (2018) [1] in the "Gran Sasso" National Park area (NPGS), in the northern part of Abruzzo region (20.1%), as well as the prevalence data generally reported in other Italian territories. In particular, from the study of Aprea et al. (2018) [1], g3c was the virus-associated subtype detected in livers from wild boars of the NPGS, which was the same subtype identified in our study. Due to the demonstration of HEV subtype 3c as the cause of infections for humans in Abruzzo [33], the abundant circulation of the same subtype in wild boars may represent an important risk for public health that needs more in-depth investigations. In particular, the authors are currently carrying on other studies that aim to obtain HEV WGS from wild animals, food and humans in Abruzzo (which is generally recognised as a hot spot Italian Region for human HEV infections). These studies are carried out with the aim to perform a phylogeny analysis that could explicitly show the role of wild animals in human diseases and the identification of the sources of infections.

The distribution of hunted wild boars is quite homogenous across the territory under study, even if slightly influenced by the orography and the degree of urbanisation of the territory (Figure 1). On the contrary, the positive results seem grouped in three main

clusters, one of which is overlapping the area with the major pig farm density. This may suggest, from one side, the presence of local risk factors in particular areas facilitating the spreading of HEV between wild boars, and, from the other side, the potential for wild boars to infect farmed pigs in the northwest part of the province under study. In these areas, a careful check of pig farm biosecurity and an improvement of sanitary surveillance and protection should be recommended.

The findings of the present study slightly differ from the results of the previous studies reported in the literature in terms of lower HEV prevalence level, but these results are still worthy of consideration. Studies conducted in some European countries showed the circulation of HEV among wild boars and pigs also in those areas. Jori et al. (2016) [34] demonstrated a high exposure to HEV in wild and domestic pigs in Corsica, confirmed also by the phylogenetic analysis. Another study conducted in Spain by Wang et al. (2019) [35] showed that HEV spreads from wild boars to pigs. A study conducted in northeast Germany in 2019, in particular, demonstrated a co-circulation of different Hepatitis E Virus Genotype 3 in pigs and wild boars [36].

Actually, HEV circulation among wild boars represents a potential risk of infection for hunters, veterinarians, operators in meat processing laboratories, slaughterers and, besides, for all consumers of raw or undercooked swine meat.

As suggested by other authors [21] studies including bio-molecular methods to detect the HEV RNA do not provide information about the virus viability, making it difficult to appropriately estimate the risk related to the consumption of pig or wild boar liver sausages or other meat products. Further studies are needed in order to investigate the virus viability to better correlate pig and wild boar meat consumption with HEV human cases in Italian regions (Abruzzo, Lazio, Campania and Calabria) where typical pig and wild boar liver products are consumed.

**Author Contributions:** Methodology, D.D., I.M., G.D.P. and C.M.; Data curation F.D.M., G.A., G.D.P. and C.M.; Formal analysis F.D.M.; Investigation S.S., C.M. and A.G.; Resources A.C., S.O. and A.D.P.; Supervision G.M.; Visualization F.P.; Writing—review & editing F.D.M., G.A., M.P. and G.C. All authors have read and agreed to the published version of the manuscript.

**Funding:** This research was funded by Ministry of Health in the framework the project—"Hepatitis E Virus: Epidemiological Surveillance and Development of WGS Diagnostic Techniques for the improvement of viral characterization, phylogeny, clustering and identification of infection sources". Internal code MSRCTE0521—VESETIS.

**Data Availability Statement:** The study did not report any data.

**Conflicts of Interest:** The authors declare no conflict of interest.

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
