# Peer review of "Detection of Hepatitis E Virus (HEV) in Pigs and in the Wild Boar (Sus scrofa) Population of Chieti Province, Abruzzo Region, Italy"

_2673-8007, doi:10.3390/applmicrobiol2040062_

Round 1
Reviewer 1 Report
Thank you for interesting article. Please add table od positive test with genotyping if It possible. Please add clear conclusion.
Author Response
Thanks for the comment, we added a phylogenetic tree that shows relationship between the isolate described and the HEV 3c Abruzzo sequences previously published (Aprea et al., 2020).
Reviewer 2 Report
I have no critical comments and questions.
I believe that the manuscript presents the results on an interesting and relevant topic.
Author Response
We thank the evaluable reviewer for appreciating our draft paper.
Reviewer 3 Report
In the submitted manuscript, the authors used RT-qPCR for the investigation of HEV prevalence in pigs and wild boars. The study shows that HEV is highly prevalent in wild boars despite its absence in pigs. Previously, several studies released to the literature, including samples from Italy, reported the circulation of HEV in domestic pigs. Although the study is iteresting, some issues still need to be addressed:
-The authors should test the plasma of animals for HEV antibodies to ensure the presence and absence of the virus in the wild boar and the pigs respectively.
- The sensitivity/limit of detection of the RT-qPCR assay should be clarified.
- Did the authors test any fecal samples for the presence of HEV RNA.
Author Response
- Thank you for your suggestion. We did not test the plasma of animals for HEV antibodies, it was a pilot study so it will be interesting to test them in the future in order to have a more complete picture.
-
Thanks for the comment.In this study the sensitivity/limit of detection of the RT-qPCR was not calculated. The method used is not yet validated nor accredited we used a cut-off of Ct 40 to gave the presence /absence. We are working on the validation tests.
-
Thank you for the comment.We did not test any fecal sample for the presence of HEV RNA, it was a study that focused the attention just on two type of matrices, liver and gall bladder.It will be interesting to test this type of sample in future.
Reviewer 4 Report
I want to thank De massis et al. for their work. They describe HEV infection in pigs family in an Italian region.
Their results are interesting and near than in other Italian regions.
I have some remarks nevertheless:
- Introduction must be improved, because, according to the genotypes, physical attempt differs: G3 is not associated with important troubles in pregnant women, such as in indian subcontinent. Other troubles, notably neurologic, are described with G3.... Furthermore, but it is just a personnal conviction, the number of deaths seems to be low (It is just a personnal remark, don't take it in account ....).
- In discussions, it would be interesting to know if in Spain, France, Germany, such results are observed.
Author Response
- Thank for your comment. We improved the introduction adding information regarding other troubles associated to HEV.
- Thank for your comment. We improved discussions adding results obrained by other European countries: spain, France and Germany.
Round 2
Reviewer 3 Report
In the resubmitted version, the manuscript has improved.
Author Response

(The authors gave the same response as above.)

Reviewer 4 Report
I want ot thank the authors for their responses and corrections.
Nevertheless, maybe my comment was not clear, but some differences exist between genotypes and clinical manifestations. For example, G3-G4 are more present in Europe since some years and can be associated with more requent extra-hepatic manifestations, less risk in pregnancy (change the text, please.), and the risk of chronicty in patients receiving immunosuppressors in case of solid transplantation.
Author Response
Thank for your comment. We improved the introduction according your suggestions.